# Modeling of Protein–Protein Interactions in Cytokinin Signal Transduction

**DOI:** 10.3390/ijms20092096

**Published:** 2019-04-28

**Authors:** Dmitry V. Arkhipov, Sergey N. Lomin, Yulia A. Myakushina, Ekaterina M. Savelieva, Dmitry I. Osolodkin, Georgy A. Romanov

**Affiliations:** 1Timiryazev Institute of Plant Physiology, Russian Academy of Sciences, Botanicheskaya 35, 127276 Moscow, Russia; hotdogue@yandex.ru (D.V.A.); losn1@yandex.ru (S.N.L.); yulia-myakushina@yandex.ru (Y.A.M.); savelievaek@ya.ru (E.M.S.); dmitry_o@qsar.chem.msu.ru (D.I.O.); 2FSBSI “Chumakov FSC R&D IBP RAS”, Poselok Instituta Poliomelita 8 bd. 1, Poselenie Moskovsky, 108819 Moscow, Russia; 3Institute of Translational Medicine and Biotechnology, Sechenov First Moscow State Medical University, Trubetskaya ul. 8, 119991 Moscow, Russia; 4Belozersky Institute of Physico-Chemical Biology, Lomonosov Moscow State University, Leninskie Gory 1, Bld. 40, 119992 Moscow, Russia

**Keywords:** cytokinin signaling, receptor, phosphotransmitter, response regulator, interaction interface, homology modeling, protein–protein interactions

## Abstract

The signaling of cytokinins (CKs), classical plant hormones, is based on the interaction of proteins that constitute the multistep phosphorelay system (MSP): catalytic receptors—sensor histidine kinases (HKs), phosphotransmitters (HPts), and transcription factors—response regulators (RRs). Any CK receptor was shown to interact in vivo with any of the studied HPts and *vice versa*. In addition, both of these proteins tend to form a homodimer or a heterodimeric complex with protein-paralog. Our study was aimed at explaining by molecular modeling the observed features of *in planta* protein–protein interactions, accompanying CK signaling. For this purpose, models of CK-signaling proteins’ structure from Arabidopsis and potato were built. The modeled interaction interfaces were formed by rather conserved areas of protein surfaces, complementary in hydrophobicity and electrostatic potential. Hot spots amino acids, determining specificity and strength of the interaction, were identified. Virtual phosphorylation of conserved Asp or His residues affected this complementation, increasing (Asp-P in HK) or decreasing (His-P in HPt) the affinity of interacting proteins. The HK–HPt and HPt–HPt interfaces overlapped, sharing some of the hot spots. MSP proteins from Arabidopsis and potato exhibited similar properties. The structural features of the modeled protein complexes were consistent with the experimental data.

## 1. Introduction

Cytokinins (CKs) are low molecular weight phytohormones that regulate a plethora of physiological processes in higher plants [1,2] (Appendix A). Plants use a modified prokaryotic two-component system (TCS), referred to as multistep phosphorelay (MSP), to transduce CK signal (Figure 1). This plant signaling system involves three main types of proteins. The first one is the multi-domain transmembrane CK receptor protein, hybrid sensor histidine kinase (HK). Its extracytosolic part facing the endoplasmic reticulum (ER) lumen or apoplast [3] represents the sensor module (SM), which includes dimerization interface, PAS (Per-Arnt-Sim) and PAS-like domains [4,5]. Transmembrane (TM) regions are represented by two to five α-helical stretches depending on receptor type [5]. The cytosolic part of the receptor includes the catalytic module, which consists of HK-dimerization (HisKA or DHpD) and H-ATPase (CAD) domains [6,7]. In addition, the receptor possesses the receiver-like (RLD or REC-like) and receiver (RD or REC) domains (considered together as a receiver module), which are located at the C-terminus of the HK protein [8,9].

The second protein type in the CK signaling pathway is the conserved histidine-containing phosphotransfer protein, or phosphotransmitter (HPt or HP), which shuttles between cytoplasm and nucleus (Figure 1). The third and the last protein in the MSP is the response regulator (RR), a transcription factor containing a conserved phosphoaccepting Asp residue [1,10,11,12].

The ligand binding in the PAS domain is assumed to be coupled with the dimerization and activation of the receptor [1,5,10,11,12]. CK signal transduction proceeds via the His–Asp phosphorelay of the prokaryotic type [13]. In the ATP-binding domain of the activated receptor, hydrolysis of ATP occurs, and the released phosphate ion binds to a conserved His residue in the HisKA domain, forming a phosphohistidine. The “hot” phosphate is then intramolecularly transferred to a conserved Asp residue in the receiver domain, which in turn interacts with the HPt, transferring the “hot” phosphate to its conserved His residue. HPt carries the “hot” phosphate to the nucleus and phosphorylates the conserved Asp residue of the RR protein. Thereafter, phosphorylated B-type RRs become able to activate the target genes [1,9,10,11,12,13].

Thus, each CK signaling step relies on the protein–protein interactions, including the dimerization of receptors, their interaction with HPt, dimerization of HPt proteins themselves, and their interaction with RRs [7,9]. Therefore, to gain insight into the molecular basis of CK signal transduction, each of the involved protein–protein interactions should be thoroughly studied. However, the 3D structure has so far been solved for only few CK-signaling elements. In such a situation, we aimed at performing a molecular modeling of the protein–protein interactions in the CK signal transduction to decipher the molecular basis of the available experimental data. Fortunately, there exist a number of proteins homologous to the components of the CK signal transduction chain. For example, several *Arabidopsis thaliana* HKs are homologous to the CK receptors AHK2–4. These proteins, unrelated to CK signaling, include AHK1 osmosensor, CKI1 and AHK5 (CKI2) proteins, and five ethylene receptors: ETR1, ETR2, EIN4, ERS1, and ERS2 (among the latter, ETR1 and ERS1 are functional hybrid HKs) [14]. Some of such protein homologs were crystallized and their full or partial structures were determined [7,9].

To date, a large number of spatial structures, determined by X-ray and/or NMR, of prokaryotic TCS and eukaryotic MSP components have been stored in the Protein Data Bank (PDB) [15]. Among them, crystal structures of the first and still unique CK receptor sensor module (AHK4sm) in complex with six different CKs provided deep insight into molecular mechanism of the CK perception [4]. The eukaryotic HK-catalytic module is represented by structures of ERS1 HisKA domain and ETR1 HATPase domain [6]. Structures of the receiver domain of three different plant HKs, ETR1 [16], CKI1 [17,18], and AHK5 (CKI2) in complex with AHP1 [19] are available. Crystal structures of a number of phosphotransmitters were also reported: *A. thaliana* AHP1 [19] and AHP2 [20], *Medicago truncatula* MtHPt1 and MtHPt2 [21], *Zea mays* ZmHP2 [22], *Oryza sativa* OsHP1 [23,24] (Appendix A). Of particular interest are structures of yeast *Saccharomyces cerevisiae* YPD1, SLN1, and SSK1 proteins. So far, the yeast MSP pathway was the only one where the three most important protein interactions were structurally characterized. The yeast MSP system consists, similarly to plants, of three components: SLN1 is a sensor hybrid HK, YPD1 is a phosphorelay intermediate, and SSK1 is a response regulator. Three types of yeast protein dimers are available in the PDB: YPD1 in complex with SLN1 receiver domain [25,26], YPD1 homodimer [27], and YPD1 in complex with SSK1 [28]. More detailed information about the available structures of MSP and TCS components and their complexes can also be found in recent reviews [7,29]. However, structural characteristics of interacting proteins and especially protein complexes critical for CK signaling are essentially missing, with very few exceptions [4,7].

Recently, using the bimolecular fluorescence complementation (BiFC) method, we have shown experimentally that each of the *A. thaliana* CK receptors AHK2–4 can interact with any of tested HPts (AHP1–3) without visible preference [30]. When studying AHK2–4 dimerization, it was shown that the receptors can form homo- and heterodimers (the latter means bound paralogs). Both AHK–AHK and AHK–AHP interactions were localized mainly to the ER. By contrast, homo- and heterodimers of various phosphotransmitters (AHPs) were found mainly in the nuclei of the analogous plant cells [30]. In the course of our parallel research, we succeeded with cloning and in depth characterizing (including structure modeling) six CK receptors (StHKs) from potato *Solanum tuberosum* [8].

The main goal for this study was to fill the gap in our knowledge concerning the structure and interaction of protein molecules directly involved in the intracellular CK signaling. Two phylogenetically distant plant species were used for the comparative analysis, *A. thaliana* (Brassicaceae) and *S. tuberosum* (Solanaceae). All important CK-signaling steps were modeled, including receptor homo- and heterodimerization, HPt homo- and heterodimerization, receptor–HPt and HPt–RR complex formation. The modeling results allowed us to reveal biochemical patterns and identify critical amino acid (aa) residues that determine the interaction specificity and interface composition of CK-signaling proteins. These critical residues termed hot spots make a large contribution to the binding affinity of protein–protein interaction [31]. In addition, the important role of transient protein phosphorylation in the modulation of protein structure and affinity needed for the intracellular CK-signaling has emerged.

## 2. Results and Discussion

### 2.1. Dimerization Interfaces of the Sensor Module

The sensor module (SM) of CK receptor is the extracytosolic part of this transmembrane protein. The crystal structure of AHK4sm is the only CK-receptor domain structure experimentally solved so far [4]. SM consists of dimerization subdomain, formed by a long (pivotal) α1-helix, which seems to be an extension of the upstream TM helix, short α2-helix and loop between these helices; large CHASE domain, including PAS and PAS-like subdomains, and short downstream linker adjacent to the downstream TM helix [5]. The SMs of AHK4 were crystallized as homodimers with different CKs bound; no apo-form structure without a bound hormone was obtained [4]. Therefore, all SM models obtained using AHK4 crystal structure as a template correspond to the ligand-bound state of the receptor, regardless of the presence or absence of ligand in the model itself.

Taking this circumstance into account, homology models of dimeric SM structures of *A. thaliana* (AHK2–4) and *S. tuberosum* monoploid var. Phureja (StHK2–4) were built (Figure 2A–D, Appendix A), and dimerization interfaces of SM homo- and heterodimers of Arabidopsis and SM homodimers of potato were analyzed (a total of nine complexes). Target proteins were at least 61% identical (Appendix A). All models had acceptable Ramachandran plot statistics (Appendix A).

Only the membrane-distal part of SM dimerization subdomain participated in the formation of protein–protein interface (PPI). The interface area of modeled complexes ranged between 946 Å^2^ and 1044 Å^2^ depending on dimer subunit composition. Most of SM interface aa were highly conserved and identical in Arabidopsis and potato (Figure 2B and Figure 3A), but there are a few variable residues on the periphery of the interface. The SM interfaces included a hydrophobic core with at least two aromatic residues (Phe and Tyr) in the center, while the interface periphery was mostly hydrophilic (Figure 2C and Figure 3B). This type of hydrophobicity spreading with a hydrophobic core and a hydrophilic rim is common for protein–protein interfaces [31]. Distribution of electrostatic potentials over the interface surfaces was also studied. This distribution had common as well as some specific traits for studied proteins (Figure 2D). For example, AHK3 had two positively charged aa, R303 and K162, in the distal part of the interface; negatively charged E182 in the proximal part; and a group of positively (R170 and R178) and negatively (D168 and E174) charged residues in the lateral area (Figure 3C). Three of these positions (R303, R170, and E174) are variable, as well as G161, which can be replaced by His in paralogous HKs (Figure 3A).

The dimerization interfaces bore 5–11 hydrogen bonds and up to five putative salt bridges according to PISA software (Appendix A). Hydrophobic *p*-values, a measure of hydrophobicity degree, differed for studied Arabidopsis (0.35–0.52) and potato (0.2–0.25) SM dimers, indicating a higher specificity of interaction in the dimers from *S. tuberosum* [32].

The calculated values of binding affinity for SM dimers ranged from −36 to −43 kJ/mol, according to Prodigy server results (Appendix A). Prodigy calculation also showed that the total number of intermolecular contacts within the threshold distance of 5.5 Å varied between 73 and 91, including 5–14 charged-charged, 12–20 charged-polar, 13–22 charged-apolar, 1–4 polar-polar, 10–14 apolar-polar and 24–26 apolar-apolar contacts.

To find critical aa (hot spots) and their interactions in the interfaces determining strength and specificity of CK receptors dimerization, virtual alanine scanning was performed for Arabidopsis and potato SM. Residues that changed the free energy ΔG of the dimer by more than 2 kJ/mol upon conversion to Ala were considered as hot spots. Robetta scanning for all the homo- and heterodimers revealed two aa positions that were hot spots in both chains of all complexes: F158 and Y175, according to AHK3 numbering. Some of the positions were hot spots in most complexes, at least in one chain: H147, K162, T171, R178 and E182 (Appendix A). For AHK3sm homodimer, additional investigation of interface residues was performed, including KFC2 and PPCheck hot spot prediction (Appendix A) and calculation of buried surface area (BSA) percent with regard to accessible surface area (ASA) (Appendix A). All of the above-mentioned positions were conserved (except K162) and confirmed (except K162 and R178) as hot spots by KFC2 calculation results for AHK3 homodimer (Appendix A). Moreover, all of these residues, except K162 and R178, had a high BSA percentage of more than 85%, according to PISA calculation (Appendix A). F158 can participate in a π–π stacking interaction with the corresponding residue of the dimer counterpart. When considering AHK3 homodimer, Y175:B (“B” after colon means subunit B) may form the hydrogen bonds with the backbone oxygen of A150:A (“A” after colon means subunit A); H147 formed stacking contacts with H147 of the dimer partner; K162 may interact with D168 of the other subunit via salt bridges and also may form hydrogen bonds with T171; R178 may form hydrogen bond with counterpart’s S156; E182 may interact with the partner’s N146 (Figure 4). Hot spots in *S. tuberosum* SM dimers were similar to the Arabidopsis ones, especially when comparing the orthologs. This is easily explained by the high identity percent between orthologous SM, 78–80% (Appendix A). In general, this also applies to bond composition formed by hot spots in potato SM dimerization interfaces. However, according to PISA analysis, the StHK2sm homodimer was distinguished by the absence of salt bridges in the PPI interface (Appendix A).

2D maps obtained using MolSurfer allowed us to investigate hydrophobicity and electrostatic potential of interfaces in the modeled MSP complexes (Appendix A). Profiles of hydrophobicity were very similar in all Arabidopsis and potato SM dimers. PQR files (see Section 3.3) were prepared for electrostatic potential determination for all dimers at two different pH values. Electrostatic potential complementarity at pH 5.5, typical for apoplast medium, was less than at pH 7.1, which mimicked ER medium ([3] and refs therein). Hence, neutral pH inside the cell should favor SM dimerization as compared to acidic pH in the apoplast. These results are consistent with the experimental data showing ER as the main compartment for CK receptor dimerization [30]. Some differences in the subunit surface complementarity were noticed between homodimers at pH 7.1 as well. Arabidopsis AHK2sm–AHK2sm and AHK3sm–AHK3sm homodimers were clearly complementary, whereas AHK4sm–AHK4sm homodimer, all heterodimers, and potato homodimers were complementary to a lesser extent. However, with any paralog combination, the complementarity areas seem to be large enough to ensure the dimer formation.

To test the assumption of the negative effect of decreasing pH on complementarity of SM dimerization interfaces, calculations with a wider range of pH values were made for a distinct complex, AHK3 homodimer. Calculations with PQR files prepared in six different pH conditions corresponding to the dispersion of pH values in the apoplast (4.5, 5.5, and 6.5) and the ER (7.0, 7.5, and 8.0) were performed. The results showed (Figure 5) a high complementarity at pH 7–8 and a presence of large non-complementary areas at pH values lower than 6.5 (especially at pH 4.5), thus confirming our assumption.

### 2.2. HisKA (DHpD) Domain Dimerization Interfaces

A set of nine complexes of *A. thaliana* and *S. tuberosum* HisKA (DHpD) domain dimers were generated with the same combinations of protein pairs as for SMs (Appendix A). HisKA domains of modeled proteins shared 35–41% sequence identity with corresponding domain of ERS1 template (Appendix A).

Unlike the SM, the HisKA domain was involved in dimerization along its entire length. Cytosolic HisKA domain consists of a long α1-helix and smaller α2-helix located in the membrane-distal part of the interface. This distal part, formed by two helices, is more conserved than the proximal part and consists mostly of hydrophobic aa. The proximal part of the dimeric interface, formed only by the α1-helix, is highly variable and involves both hydrophilic and hydrophobic residues (Figure 6).

Interaction interfaces of the HisKA domain of the receptor covered a much larger area than the other interfaces considered here. Contact area of the investigated HisKA domain interfaces ranged between 1977 and 2267 Å^2^. HisKA domain dimers were twist-shaped and their interfaces included conserved and variable parts. At the same time, the number of hydrogen bonds was smaller than in the other interfaces (5–8 depending on subunit).

The range of interaction energy values of HisKA domain dimers, determined by Prodigy, was between –40 and –57 kJ/mol (Appendix A). Total number of intermolecular contacts ranged between 128 and 151, including 8–14 charged-charged, 3–6 charged-polar, 25–34 charged-apolar, 2–6 polar-polar, 11–27 apolar-polar and 65–88 apolar-apolar contacts.

Thus, substantial differences between SM and HisKA dimerization properties were revealed. HisKA dimers had larger interface area and higher average interaction energy. The number of charged-charged and polar-polar contacts was pretty similar and the number of charged-polar contacts in HisKA dimers was even less than in SMs despite a much higher total number of contacts in HisKA. At the same time, apolar-apolar contacts in some cases amounted to more than half of the total contacts in HisKA complexes, whereas, in SM dimers, this proportion was much lower. Differences were also evident in the hydrophobicity pattern: HisKA dimers had a large hydrophobic zone in the distal part of the interface, whereas in SM dimerization interfaces, hydrophobic and hydrophilic areas were distributed fairly evenly. Interestingly, among all studied MSP dimers, the largest number of intermolecular contacts and the highest value of the interaction energy were inherent to HisKA and SM homodimers of the same protein, AHK3. Hydrophobic *p*-value of HisKA dimers (ranged between 0.06 and 0.25) was lower than that of SM dimers (0.20–0.52), making evident the higher hydrophobicity of the HisKA complex interface.

Despite the fact that HisKA dimers had large interface areas and a lot of intermolecular contacts, Robetta alanine scanning revealed only a small number of hot spots in these complexes (Appendix A). There were no residue positions that were detected as presumable hot spots for all complexes. AHK3 homodimer had only two hot spots detected in chain (subunit) A and three ones in chain (subunit) B. K451 and Q485 positions, which were hot spots in both chains of AHK3 HisKA homodimers, were also hot spots in a few other complexes. Only K451:A of AHK3 HisKA homodimer was confirmed as a hot spot by PPCheck and KFC results; in addition, unlike Q485, K451 residue was shown to be very conserved (Appendix A). *S. tuberosum* HisKA dimers differed from the Arabidopsis ones in alanine scanning. For example, eight hot spot positions were detected for chain B of StHK3 HisKA homodimer; this was much more than in other complexes. Three of these positions were unique for this dimer: F407, D469 and L489 (corresponding to AHK3′s I420, D482 and L502, respectively).

MolSurfer maps for hydrophobic and electrostatic complementarity of interfaces were obtained using PQR files prepared for pH 7.3, close to that in the cytosol (Appendix A). Besides hydrophobic complementarity, all of the HisKA dimeric interfaces were complementary in electrostatic potential as well. These observations were equally true for the Arabidopsis and potato CK receptors. Thus, the possibility of CK receptors to form heterodimers (consisting of paralogs) as well as homodimers was confirmed.

### 2.3. Receptor–Phosphotransmitter Interactions

Homology models for all combinations of complexes of Arabidopsis phosphotransmitters AHP1–3 bound to receiver domains of Arabidopsis CK receptors AHK2–4 were built, and the AHK5rd–AHP1 model served as a control (compared to the AHK5rd–AHP1 crystal structure, PDB ID: 4EUK). A total of 10 *A. thaliana* complexes were modeled (Figure 7, Appendix A). Receiver domains of *S. tuberosum* CK receptors StHK2–4 were modeled as complexes with StHP1a phosphotransfer protein, a total of three complexes (Figure 7, Appendix A).

The structures of the receiver domains of CK receptors include a five-stranded parallel β-sheet surrounded by five main (and often one additional) α-helices. Phosphotransfer proteins from Arabidopsis comprise six α-helices and lack β-strands. In the complex, the α1 helix of the receiver domain was the closest to the phosphotransmitter domain, forming a prevailing part of the receptor–phosphotransmitter interface. The phosphorylated Asp residue (D941 in AHK3rd) resided in the other region of the AHKrd molecule, namely, at the edge of the β3 strand adjacent to the loop L5. This site was accessible for the phosphoaccepting His residue (H82 in AHP2) of the phosphotransmitter. In the latter, three α-helices (α2, α3 and α4) of a total of six were involved in the formation of the interaction interface.

According to the PISA assessment, the modeled MSP complexes differed in interaction interface properties (Appendix A). Most of the complexes had hydrophobic *p*-values in the range between 0.27 and 0.81, with AHK3rd–AHP1 complex distinguishing by the low *p*-value. Interface area of different complexes ranged between 777 and 942 Å^2^. Number of hydrogen bonds in the interaction interfaces of modeled complexes varied from 5 to 13, and number of salt bridges varied from 1 to 9. Predicted binding affinity according to Prodigy calculations ranged between −40 and −50 kJ/mol (Appendix A).

Almost half of HKrd and most HPt interface aa residues were highly conserved (ConSurf scores of 7 and above) (Figure 7B and Figure 8A). Notably, the interface region is the most conserved part of both interacting proteins. Distribution of hydrophobic and hydrophilic regions on the interfaces of all the investigated HK–HPt complexes was very similar. There was a hydrophobic core in HKrd and HPt interfaces, surrounded by hydrophilic residues with an additional small hydrophobic area on the periphery (Figure 7C and Figure 8B). The surface patterns of electrostatic potential showed complementarity of the HKrd and HPt interaction interfaces. The central region was neutral or almost neutral in both proteins with a clearly negative sector on one edge of the HPt interface, which matched a positive area of the HKrd interface and positive sector on another edge, which matched a negative part of the HKrd interface (Figure 7D and Figure 8C). According to MolSurfer results, PPI interfaces of all Arabidopsis and potato complexes had very similar and complementary patterns of hydrophobicity. Electrostatic potential complementarity at pH 7.3 (cytosolic pH) was clearly visible in all combinations of dimer counterparts, with StHK2rd–StHP1a and StHK3rd–StHP1a complexes distinguished by the perfect matching (Appendix A). This high electrostatic complementarity as well as a high level of conservation of the interface residues (especially in HPt counterpart) can explain HK–HPt interaction promiscuity shown in our experiments earlier [30].

To reveal critical aa and their interactions in the interfaces determining binding between CK receptors and phosphotransfer proteins, virtual alanine scanning was performed (Appendix A, Appendix A). All 13 complexes were investigated not only to characterize individual pairs of proteins, but also to highlight general trends. For the HKrd counterpart, three hot spots’ positions K1013, N898, and N901 (AHK3 numbering) were clearly revealed in all the studied complexes, and two additional putative hot spots, V900 and R903, were uncovered in more than two, but not in all complexes. For HPt, two strongly conserved hot spots were revealed, Q83 and S87, as well as two less conserved ones, D54 and S90 (AHP2 numbering).

For AHK3rd–AHP2 complex, additional investigation of the interface residues was performed, including KFC2 and PPCheck hot spot prediction (Appendix A), and calculation of BSA percent with regard to ASA (Appendix A). The hot spot status was confirmed for N898, N901, V900, and R903 positions in AHKrd counterparts, but not for K1013. Positions N898, N901, and K1013 were highly conserved (ConSurf scores 7, 8, and 9, respectively); N898, N901, and V900 were characterized by more than 90% BSA percentage, whereas K1013 had as little as ~40%. All the predicted hot spots of HPt interface were highly conserved (ConSurf scores 8 to 9), but only S90 was confirmed as a hot spot by the KFC2 and PPCheck services. For S90, BSA was almost 100%, for Q83 and S87 above 80%, while for D54 only about 60%.

K1013 did not form any intermolecular hydrogen bonds or salt bridges, but its intramolecular salt bridge with D941 is well known to stabilize active conformation and provide Mg^2+^ binding needed for phosphorylation [17]. N898 and N901 of AHK3 formed at least three hydrogen bonds with side chains of AHP residues. N898 interacted with Q83 and S87 of HPts, N901 formed a hydrogen bond with S90 (Appendix A, Figure 9). These Asn residues are located within a L1–α1 helix stretch in the RD. In the HPt counterparts, Ser and Glu residues forming hydrogen bonds with Asn residues of HKrd are located in the α4 helix.

Most results of alanine scanning were similar between Arabidopsis and potato, except minor features. For example, K1142 (corresponding to K910 in AHK3) was detected as hot spot only in the RD counterpart of StHK2rd–StHP1a complex, while the other complexes had no hot spot in this position.

Pekárová et al. [17] studied the interaction between CKI1 (which is a histidine kinase, but not a cytokinin receptor) and AHP1–6 proteins. Results of their BiFC study suggested a tight interaction of CKI1 with AHP2, AHP3 and AHP5, a weaker interaction with AHP1, and no interaction with AHP4 and AHP6. Yeast two-hybrid assay experiments confirmed these results except that there was no interaction between CKI1rd and AHP1, and CKI1rd–AHP5 interaction was weaker than the interactions with AHP2 and AHP3. ELISA experiments also showed strong interaction of CKI1rd with AHP2 and AHP3, while the interaction of CKI1rd with AHP5 was weaker. In the case of AHP2 and AHP3, these data are consistent with our results. The difference in the case of AHP1 can be explained both by different localization and structural features of CKI1 and CK receptors. CKI1 is located in the plasma membrane [17], while CK receptors are localized mainly in the ER [3,30]. The main structural difference in the cytosolic parts of these proteins is the absence of a receiver-like domain in CKI1, as well as in all histidine kinases except CK receptors. Hypothetically, this domain may play an indirect role in the interaction with HPt. There was also a difference in the RD–HPt interaction interface upon comparison of the CKI1 and CK receptors. N901 (AHK3 numbering), which was a hot spot in our calculations, was replaced by a serine in CKI1 (S997, CKI1 numbering). This may lead to weaker interactions with some HPts.

Bauer et al. [19] investigated the interaction between CKI2 (AHK5) and AHP1–6 proteins. It was shown in BiFC experiments that CKI2 interacted with all the HPts except AHP4. Interactions with AHP2 and AHP5 were a bit more pronounced than the others. Binding affinities of CKI2 for AHP1–3 were very similar, according to surface plasmon resonance, with a small decrease of the dissociation constant for interaction with AHP2. This correlates with our data on the promiscuous interaction of AHK2–4 with AHP1–3. It is also notable that N901 (AHK3 numbering) was not replaced in CKI2 (N789, CKI2 numbering), unlike in CKI1. This may indicate the effect of this aa position on the specificity of RD–HPt interaction.

### 2.4. HPt–HPt Interface and Its Comparison to the HPt–HKrd Interface

Two structures of phosphotransfer proteins in dimeric form were found in the PDB: OsHP1 (PDB IDs: 1YVI, 2Q4F) and YPD1 (PDB ID: 1C02) homodimers. YPD1 has only about a 20% sequence identity with HPts from *A. thaliana* and *S. tuberosum*. Thus, it cannot be a relevant template. OsHP1 structure contains two identical chains (A and B), corresponding to two HPt subunits. To check whether PDB dimer of OsHP1 has a biological meaning, we applied two protocols. At first, the structure was verified using “Dimer Classification” option in a ClusPro service. The principle of this procedure is blind redocking of dimer subunits. The docking results showed that this structure, indeed, should be a dimer with likely natural (biological) conformation. The probability of being a natural dimer was 80%, so the model based on this structure should be biologically relevant (Appendix A).

The second step was the blind docking of AHP2 homology model monomers using ClusPro and PatchDock servers. Results were compared with the OsHP1 crystal structure and the homology model of AHP2 dimer based on OsHP1 structure (PDB ID: 1YVI). The best result of PatchDock docking, the structure with the lowest atomic contact energy (ACE), was chosen from top 20 structures ranked by PatchDock main score. Both ClusPro and PatchDock best models had a configuration very close to OsHP1 crystal structure and homology model of AHP2 dimer (Appendix A). All-atom RMSD (root-mean-square deviation of atomic positions) for alignment between OsHP1 crystal structure and ClusPro AHP2 homodimer was 1.115 Å and 0.723 Å for alignment between OsHP1 crystal structure and PatchDock AHP2 homodimer, in the case of AHP2 homodimer homology model aligned with ClusPro and PatchDock dimers it was 1.768 Å and 1.375 Å, respectively. In this conformation of HPt-homodimer and HPt–HKrd heterodimer, the interfacial surface areas of HPt largely overlapped (Figure 10).

Sequence identity between modeled proteins and OsHP1 template ranged from 43% to 51% (Appendix A). A total of seven HPt–HPt dimer models were built: *A. thaliana* AHP1–3 homo- and heterodimers and *S. tuberosum* StHP1a homodimer (Appendix A, Figure 11, Appendix A).

According to the PISA interface summary, 25 residues of AHP2 were involved in the interaction with AHK3 receiver domain, and 36 in homodimerization (some of them were specific for A or B chain). Both interactions shared 19 residues (Figure 10B). The interface area and number of interface hydrogen bonds and salt bridges were determined by PISA as follows: 818 Å^2^, 13, and 6 for AHK3rd–AHP2 complex, and 1150 Å^2^, 12, and 3 for AHP2–AHP2 dimer. Complexes were also checked in Prodigy service with default temperature setting (25 °C) resulting in the binding energy of AHK3rd–AHP2 complex as -33.47 kJ/mol, and for AHP2–AHP2 dimer as −48.53 kJ/mol. Thus, both dimeric structures were biologically relevant, and AHK3rd–AHP2 complex seems to be less stable than AHP2–AHP2 dimer.

Alanine scanning of HPt–HPt complexes showed that the main hot spots were Q83 and D54 (AHP2 numbering), located at the center of the interaction interface (Appendix A). Thus, at least one hot spot position was the same as in the AHKrd–AHP interaction. However, the trends in HPt–HPt interaction were not as unequivocal as in HKrd–HPt interaction, and hot spot positions could vary depending on complex composition and protein sequence. Particularly, in AHP2 homodimer (Appendix A), four hot spots were identified by Robetta in the chain A, D54, Q76, Q83, and S87, and two of them, D54 and Q83, were confirmed by KFC2, but only one, Q83, was also confirmed by PPCheck. Within the chain B, two hot spots were detected by Robetta: Q46 and Q83, both confirmed by KFC2, and Q83 additionally confirmed by PPCheck. D54 and Q83 in the chain A, as well as Q46 and Q83 in the chain B, have BSA percentage of more than 70% (Appendix A). The following residues were suggested to form hydrogen bonds in AHP2 homodimer (Figure 12, Appendix A): D54:A with Q83:B; Q76:A with Q46:B; Q83:A with side chain of D54:B and the backbone oxygen of L50:B, Q46:B with S75:A, besides the interaction with Q76:A. A set of hot spots in *S. tuberosum* StHP1a homodimer was substantially different from that in Arabidopsis HPt homodimers. D52 and Q80 (corresponding to D54 and Q82 in AHP2) were no longer hot spots in both StHP1a chains, whereas F41 and E44, corresponding to F43 and E46 in AHP2, were hot spots only in StHP1a. In other respects, the structures of potato and Arabidopsis HPt dimers were very similar.

MolSurfer results showed that HPt–HPt complexes in the chosen conformation demonstrate only a moderate residue hydrophobic complementarity, but that of atomic hydrophobicity was rather high, similarly to the complementarity of electrostatic potential (Appendix A).

### 2.5. RRrd–HPt Interface and Its Comparison to AHP–AHKrd Interface

Comparison of YPD1–SLN1 and YPD1–SSK1 crystal structures showed a marked similarity (Appendix A) of HK receiver domain and RR conformations when forming a complex with phosphorelay intermediate. Thus, AHK5–AHP1 complex remained the main template for RRrd–HPt complexes modeling, and the second template for the RD was the *E. coli* CheY receiver domain.

*A. thaliana* ARR 1, 2, 10, and 11 were modeled in complex with AHP2, whereas *S. tuberosum* StRR 1a and 11 were modeled in complex with StHP1a (a total of six complexes) (Appendix A).

The interface part of RRrd surface was more variable compared to HKrd (Figure 13). RRrd interfaces have similar charge distribution over the surface, but positive-charged area was more pronounced, especially in ARR11 and StRR11. On the contrary, hydrophobic regions were less pronounced and had no clear boundaries. RRrd–HPt interfaces comprised an area varying between 881 and 1013 Å^2^, which was a bit larger than that in the HKrds–HPts. The interfaces included up to nine hydrogen bonds and up to 10 putative salt bridges.

Alanine scanning highlighted three positions, D45, T47, and K138 (ARR1 numbering) that might be considered as hot spots in at least three RRrd–HPt complexes (Appendix A). Surprisingly, ARR1 itself had only T47 and K138 as hot spots. HPt counterpart had two hot spot positions in RRrd–HPt complex: L35 and S87 (AHP2 numbering). Both T47 of ARR1 and S87 of AHP2 had BSA of 100% in the ARR1rd–AHP2 complex (Appendix A).

Comparing the main HKrd hot spots of with residues at the same positions of RRrd, we can see that one HKrd’s Asn (N898, AHK3 numbering) was replaced by Asp (D45 in ARR1), and another HKrd’s Asn (N901, AHK3 numbering) was replaced by Cys (C48 in ARR1) in all the studied RRs, except ARR11 and StRR11, where Asn901 changed to Trp.

*S*. *tuberosum* complexes differed from Arabidopsis ones in alanine scanning results (Appendix A). RD counterpart of StRR1a(rd)–StHP1a had the largest number of hot spots compared to the other studied RRrd–HPt complexes, and included a unique I43 residue (corresponding to I51 of ARR1). S87 in HPt counterpart (corresponding to S90 of AHP2) was a hot spot only in this protein pair. StRR11(rd)–StHP1a had two unique hot spots: R127 (corresponding to R141 of ARR1) in the RD and Q26 (corresponding to Q28 of AHP2) in HPt part.

MolSurfer results showed a high level of hydrophobic and electrostatic complementarity for all the studied RRrd–HPt complexes (Appendix A).

The occurrence of most RRrd–HPt interactions modeled in this study was consistent with experiments by Dortay et al. [33].

Verma et al. [34] investigated putative key aa in ARR4-AHP1 interaction. Using homology modeling, they have suggested that five ARR4 aa (D45, R51, Y96, C97 and P148) could play a critical role in ARR4–AHP interaction. Two of these residues, D45 and Y96, were confirmed experimentally as interaction determinants. There are several reasons why these results are not related to our identification of RRrd–HPt hot spots. First of all, RRrd interaction interfaces include quite extensive variable zones where properties may vary significantly depending on the specific protein. Second, ARR4 belongs to A-type ARRs, whereas our study considered B-type ARRs. Finally, as a criterion for the selection of key aa, Verma et al. [34] used the number of intermolecular bonds formed by these residues but not the interaction energy.

### 2.6. Effect of Phosphorylation on the HKrd–HPt Interactions

To study the effect of specific phosphorylation on HKrd–HPt interaction, AHK3rd–AHP2 dimer models were built using options of residue phosphorylation and Mg^2+^ ion presence. Phosphoaccepting residues have been modified in Vienna-PTM service. His was virtually phosphorylated at Nε atom, with –2 charge (without protonation), and the same charge was used for phosphoaspartate (Appendix A). Five versions of the complex were analyzed, one Mg^2+^-free wild type and four with bound Mg^2+^: wild type, dimer with phosphorylated D941 in AHK3rd, dimer with phosphorylated H82 in AHP2, and dimer with both phosphorylated phosphoaccepting residues. Additionally, two dimers with phosphorylation mimicking mutations in AHK3rd (D941E) and in AHP2 (H82E) were modeled, also in the presence of Mg^2+^ (Table 1, Appendix A).

Earlier, Scharein and Groth [35] in their experimental research of interaction between AHP1 and ethylene receptor ETR1 (ethylene sensing HK structurally close to CK receptors) made the mutants of these proteins mimicking either permanent phosphorylation or preventing phosphorylation. The affinity of interaction decreased when both partners were either in non-phosphorylated or phosphorylated state. However, when only one of the interacting proteins was phosphorylated, the high binding affinity was restored.

Pekárová et al. [17] experimentally showed that adding Mg^2+^ to the CKI1rd–AHP2 complex increased binding affinity, but addition of Mg^2+^ together with BeF_3_^-^ reduced it to a level lower than that of the apo-forms. In case of CKI1rd–AHP3 complex, adding Mg^2+^ led to a slight decrease in affinity, whereas addition of Mg^2+^ together with BeF_3_^-^ increased the affinity to a level higher than that of the apo-forms.

Our analysis of AHK3rd–AHP2 dimer models in different phosphorylation states (Table 1) showed that Mg^2+^-bound MSP partners markedly increased their mutual affinity in comparison to the apo-form. A further affinity increase was obtained when D941 of AHK3rd was shifted into the phosphorylated state in the presence of magnesium ions. In the opposite case, when AHP2 alone was phosphorylated (H82 transformed to phosphohistidine, also in the presence of Mg^2+^), the affinity of RD–HPt binding dropped to a level even lower than that of the apo-forms. Phosphorylation of both phosphoaccepting residues led to an affinity decrease compared to Mg^2+^-bound non-phosphorylated form, although the affinity remained slightly higher than that of the apo-form.

Phosphorylation-mimicking mutants of AHK3rd (D941E) and AHP2 (H82E) in the presence of Mg^2+^ showed the affinity higher than Mg^2+^-free wild type form. However, in comparison with Mg^2+^-bound wild type, results were qualitatively consistent (although less pronounced) with the data for phosphorylated MSP proteins: D941E mutation slightly increased the affinity while H82E mutation decreased it.

The effect of the phosphorylation on binding affinity can be explained by changes in the surface electrostatic potential (Appendix A). Phosphorylation of D941 in AHK3 enlarged negatively charged zone of the receiver domain, whereas phosphorylation of H82 in AHP2 changed the charge of corresponding interface area from positive to negative, thus leading to a reduction of electrostatic complementarity and decrease in affinity. These affinity changes have a clear biological meaning: HPt has to dissociate from receptor after H82 phosphorylation to start moving into the nucleus.

## 3. Materials and Methods

### 3.1. Homology Modeling

SWISS-MODEL web service has been used for template search in homology modeling [36]. Sequence alignments were performed with NCBI BLAST [37] and Clustal X 2.1 [38]. Alignment files for individual models are provided in Appendix A. Template assignment for the models is given in Table 2 and Appendix A [4,6,17,19,23,24,39]. To compensate for low sequence identity between templates and target proteins, to avoid incorrect geometry, and also to unify the modeling protocols, all HKrd-HPt and RRrd-HPt complexes were modeled using the multitemplate algorithm.

Modeling of the *A. thaliana* and *S. tuberosum* (double monoploid var. Phureja) protein structures was accomplished in Modeller 9.20 [40] using an *automodel* class for comparative modeling. For each protein, 200 models were built, and the best model was selected according to the value of DOPE (discrete optimized protein energy) score [41] calculated by Modeller.

The main set of HKrd–HPt complexes was built in the absence of Mg^2+^ ions to make the resulting model files compatible with the wide range of software. An additional AHK3rd–AHP2 model was built in the presence of Mg^2+^ for further modification aiming to study the phosphorylation effect. Models built and studied in this work are available in Appendix A.

### 3.2. Structure Optimization, Validation, and Modification

Phosphoaccepting residues were modified in Vienna-PTM web server [42]. Histidines were virtually phosphorylated at the Nε atom. Modification procedures were performed on selected models before minimization.

All the best models underwent minimization. After adding hydrogen atoms, the models were energy minimized in UCSF Chimera 1.13.1 [43] using an AMBER ff14SB force field [44] with 300 steps of steepest descent and 300 steps of conjugate gradient; step size was 0.02 Å in both cases.

Stereochemical quality of the models was assessed with ProCheck [45] implemented in PDBsum Web service [46], ProSA-web [47], and QMEAN server [48]. The models obtained had acceptable Ramachandran plot parameters—after minimization, the percentage of residues in the most favored regions was at least 87%, and percentage of residues in disallowed regions did not exceed 1.2%. Minimization procedures increased a percentage of residues in the additional allowed regions and slightly decreased their number in the most favored regions in the Main Ramachandran plot statistics of models, but fixed most of the distorted geometry, including unusual bond lengths and angles.

In some cases, models were refined using UCSF Chimera: distorted geometry was manually fixed by adjusting bond angles and lengths to their “ideal” values according to ProCheck; side chains orientation was refined with Dunbrack rotamer library [49].

### 3.3. Interface Properties Investigation

Alanine scanning was performed with Rosetta implemented in Robetta [50] with default settings. Hot spots (positions, substitution of which led to the largest change of free energy of interaction) were identified according to the ΔΔG values of complex formation after virtual mutation of a single aa residue to alanine. Positions with ΔΔG of 2 kJ/mol or greater were considered as hot spots. Mitchell Lab KFC2 server [51] and PPCheck server [52] were used for additional prediction of hot spots.

Visualization and superposition of the models were accomplished with UCSF Chimera. Interfaces parameters, including interface area, ΔG (solvation energy gain upon interface formation), total binding energy of the proteins, hydrophobic *p*-value and information about hydrogen bonds, salt bridges, and disulphide bonds formed between the interfacing chains were obtained using QtPISA tool [31,53]. PRODIGY server with the default 25 °C temperature setting was used to explore binding affinity and dissociation constants of the MSP complexes [54].

To study hydrophobicity and electrostatic potential of interfaces, 2D maps were calculated using MolSurfer service [55]. Electrostatic potential was calculated with default settings: protein dielectric constant: 4; solvent dielectric constant: 80; ionic strength 150 mM; ion radius: 1.5 Å; probe radius for generating molecular surface: 0.0 Å (0.0 Å here resulted in using van der Waals surface to separate the protein and solvent). PQR files were generated with PDB2PQR server version 2.1.1 [56] with AMBER force field chosen for calculations. PROPKA was used to assign protonation states at provided pH. pH values were set different in various complex types, depending on their localization: 7.3 for cytosol (HisKA domains dimers, HKrd–HP complexes), 7.2 for nucleus (HP dimers and RRrd–HP complexes); for sensor module dimers, two types of calculations were performed, with pH 7.1 corresponding to ER lumen value, and pH 5.5, as the average apoplast value ([3] and the references therein). Additionally, a study for AHK3 sensor module homodimer was conducted with a wider range of pH values: 4.5, 5.5, 6.5, 7.0, 7.5, and 8.0.

### 3.4. Surface Coloring

Conservation analysis along with corresponding coloring was accomplished with ConSurf server [57,58]. The HMMER homolog search algorithm with one iteration and 0.0001 E-value cutoff and CLEAN_UNIPROT Proteins database were used. In addition, 150 sequences were retrieved with identities of no less than 35% and no more than 95%. The CLUSTALW method was used to build the multiple sequence alignment. The Bayesian calculation method and default evolutionary substitution model (best model) were used. Models obtained after ConSurf calculation were colored according to sequence conservation in UCSF Chimera.

Surfaces were colored by hydrophobicity with UCSF Chimera “kdHydrophobicity” attribute according to the hydrophobicity scale of Kyte and Doolittle [59].

Surface coloring by electrostatic potential was performed also in UCSF Chimera, using Coulombic surface coloring, with a range between −10 and 10 kcal/(mol·e). Distance-dependent dielectric option was set on “true” with dielectric constant set on default 4.0 and distance from surface set on default 1.4 Å.

### 3.5. Protein–Protein Docking

Blind protein–protein docking was performed in ClusPro [60] and PatchDock [61]. Clustering RMSD in PatchDock was 4.0 and complex type was set to default. ClusPro has been launched with default settings. ClusPro was also applied to check the template structures’ possibility of being a biological dimer (not a crystallographic artifact), using the “Dimer Classification” option in this service.

## 4. Conclusions 

In this article, we present an in silico investigation of all types of protein–protein interactions involved in CK signaling. We used programs that were widespread and proved to be reliable for the required calculations [31,62]. The ability of CK receptors to give rise to heterodimers (made up of the paralogs) together with homodimers in vivo was consistent with a high conservation of interface residues, especially in the sensor module dimerization interface, and also with a high level of complementarity in hydrophobicity and electrostatic potential. The latter complementarity was assessed as strong at quasi-neutral pH but became less pronounced at acidic pH (5.5) close to that in the apoplast. Thus, previously obtained experimental results showing that receptor dimerization occurred mainly in the ER were corroborated and explained, at least partly. By means of a virtual alanine scanning, a number of aa were identified as hot spot residues for sensor module dimerization interface. Two of these residues (F158 and Y175, AHK3 numbering) behaved as hot spots in all of investigated complexes. Only minor differences in dimerization interface properties were observed between Arabidopsis and potato receptor dimers. The promiscuity in interactions between AHK2–4 receptors and AHP1–3 phosphotransmitters was also explained by strong conservation of the interface (especially in phosphotransmitter counterpart) and high electrostatic complementarity of interfaces of all studied complexes. Three hot spot positions (N898, N901, and K1013, AHK3 numbering) were predicted for the receptor interface in HK–HPt complexes. Two of them (N898 and N901) formed hydrogen bonds with the phosphotransmitter hot spot counterparts (Q83, S87 and S90, AHP2 numbering). Models of HPt–HPt complexes, based on the OsHP1 template, showed a large number of residues in the dimerization interface also involved in the interaction with receptor’s RD. Conformation of HPt complex with RR receiver domain was similar to a complex with receptor’s RD, but hot spot positions were found to be different. We also showed that the calculated affinity of HKrd–HPt complexes depends on the presence of Mg^2+^ ion and on the phosphorylation state of conserved residues (His, Asp). Taking into account that CK signaling is a phosphorelay based on protein recognition and tight interaction, where the “hot” phosphate residue serves as a “transmissible baton”, this specific dependence of affinities of interacting proteins on their phosphorylation state has a clear biological meaning. To conclude, our MSP protein modeling and computational model analysis not only explained recent experimental results but also provided justified predictions that can build up the basis for CK signaling modification by site-directed mutagenesis. In addition, the predicted hot spots can serve as targets for the development of new potato varieties. It is known that CKs play an important role as part of the regulatory hormonal complex in the formation of potato tubers [63]. Therefore, engineering of proteins associated with the MSP pathway may be beneficial for potato breeding.

## Figures and Tables

**Figure 1 ijms-20-02096-f001:**
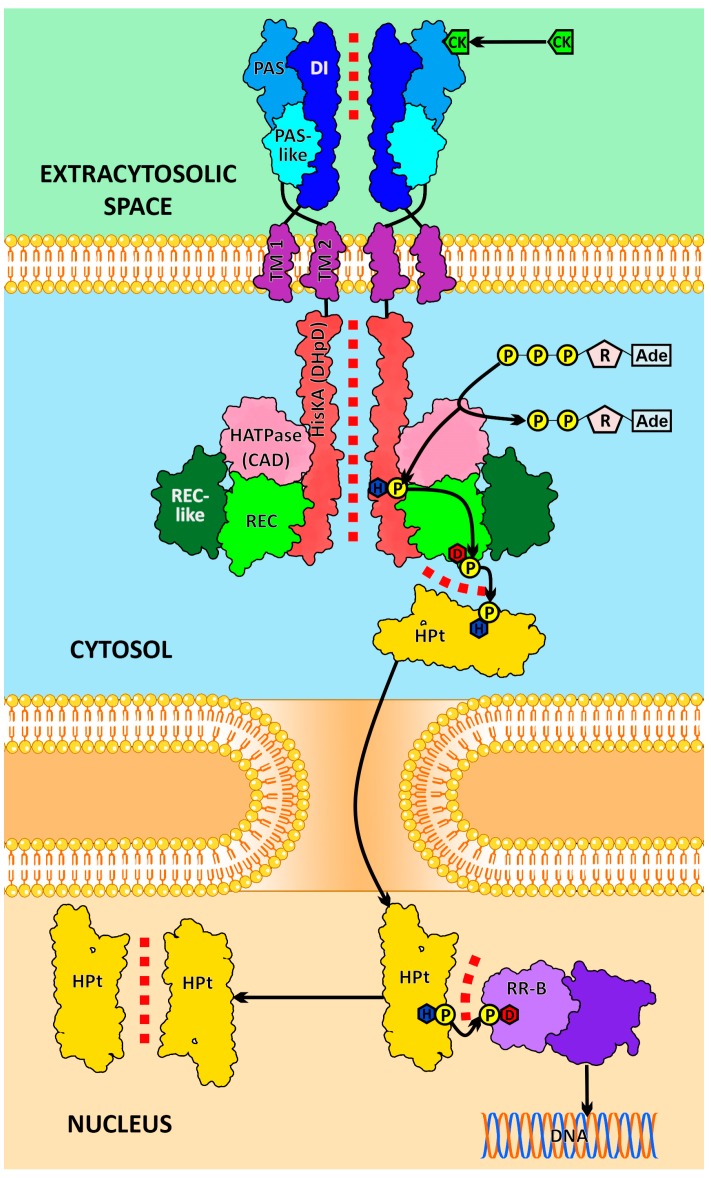
Intermolecular interactions in the cytokinin signaling pathway. CK, cytokinin; Ade, adenine; R, ribose; P, phosphate; D, conserved aspartate; H, conserved histidine; DI, dimerization interface domain of the sensor module; PAS and PAS-like are subdomains of the CHASE domain of the sensor module; TM1 and TM2, transmembrane domains; HisKA (DHpD), histidine kinase A domain (dimerization and histidine phosphotransfer domain); HATPase (CAD), adenosine triphosphatase domain (catalytic and ATP-binding domain); REC-like, receiver-like domain; REC, receiver domain; HPt, histidine-containing phosphotransfer protein (phosphotransmitter); RR-B, type B response regulator (transcription factor). Protein–protein interactions (PPI) are indicated by a red dotted line.

**Figure 2 ijms-20-02096-f002:**
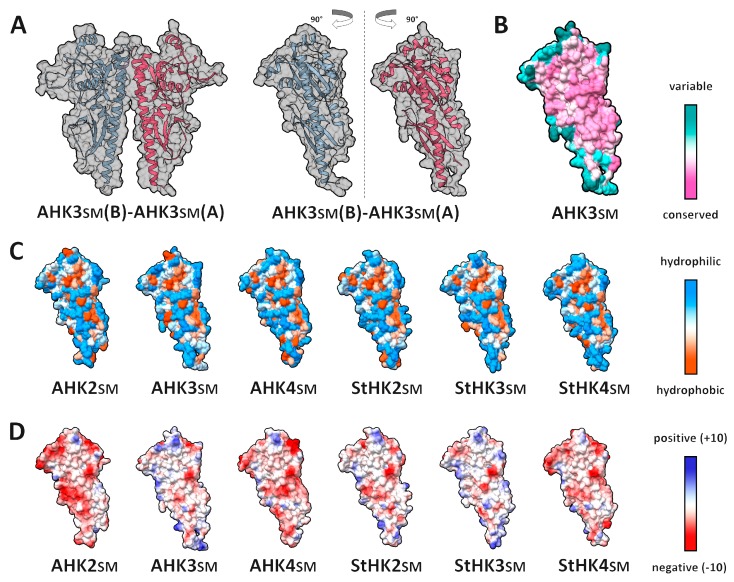
Properties of CK receptor sensor module surfaces forming protein–protein interaction interfaces. (**A**) overall view of AHK3sm homodimer (left) and its rotated subunits showing interface side (right); (**B**–**D**) interface side of the modeled SM surfaces colored by different attributes: (**B**) conservation; (**C**) hydrophobicity; (**D**) electrostatic potential.

**Figure 3 ijms-20-02096-f003:**
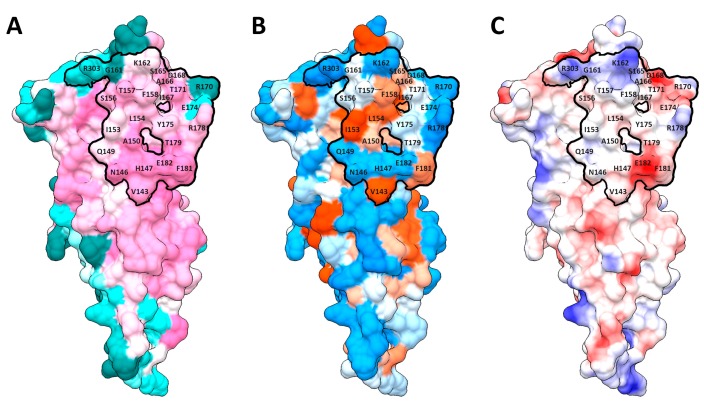
Properties of PPI interface side of AHK3 homodimer sensor module surface, with highlighted dimerization interface. (**A**) conservation; (**B**) hydrophobicity; (**C**) electrostatic potential. Color legends are the same as in Figure 2.

**Figure 4 ijms-20-02096-f004:**
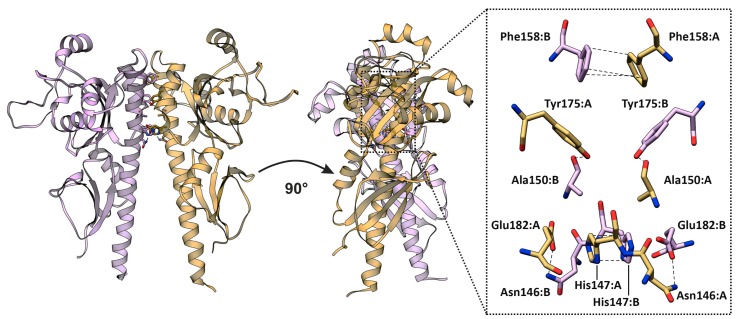
Interactions formed by the main hot spots in the dimerization interface of AHK3 sensor module homodimer. Different colors denote two subunits of the homodimer: chains A and B colored in purple and hazel, respectively.

**Figure 5 ijms-20-02096-f005:**
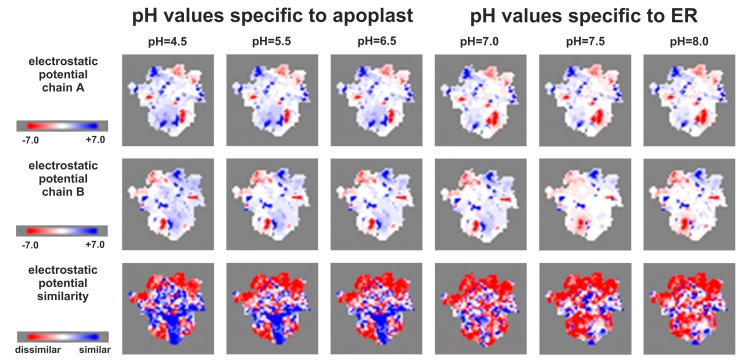
2D maps (projections) of electrostatic potentials of AHK3 sensor module homodimer interface, calculated with MolSurfer server, using PQR files prepared in different pH conditions. Top row is the projection of chain A, middle row—chain B, and bottom row is the electrostatic potential similarity of both chains. For electrostatic potentials blue color means positive charge (+7), white—neutral (0), and red—negative (−7). For electrostatic potential similarity: blue—most similar zones, red—most dissimilar zones. Greatest dissimilarity (red) means greatest electrostatic potential complementarity.

**Figure 6 ijms-20-02096-f006:**
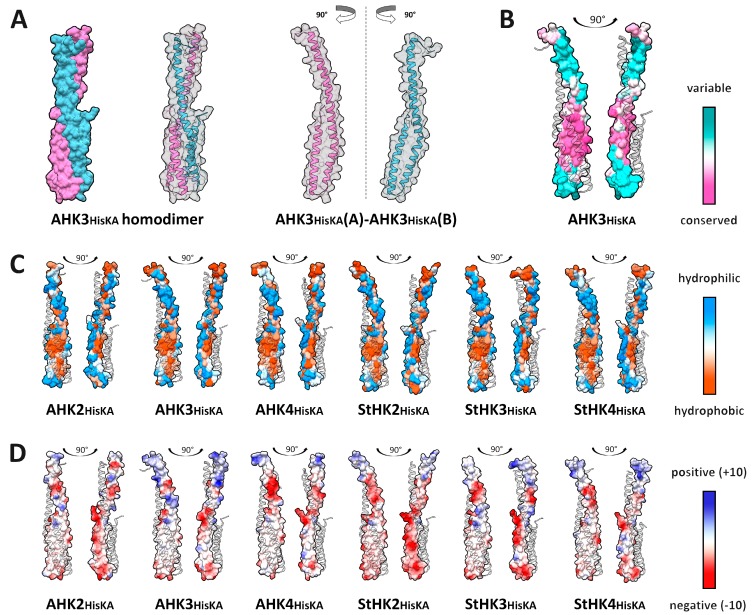
Properties of HisKA domain interaction interfaces of CK receptor. (**A**) overall view of AHK3 HisKA domain homodimer with opaque (left) and transparent surface (center) and its rotated subunits showing interface side (right); (**B**–**D**) interfaced side of the modeled HisKA domains surfaces colored by different attributes: (**B**) conservation; (**C**) hydrophobicity; (**D**) electrostatic potential.

**Figure 7 ijms-20-02096-f007:**
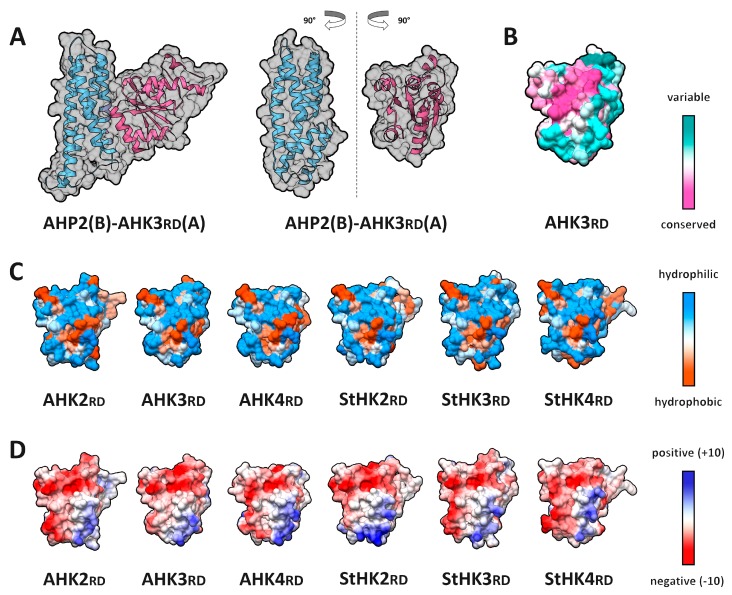
Properties of interaction interfaces of CK receptor receiver domains. (**A**) overall view of AHK3rd–AHP2 complex (left) and its rotated subunits showing interface side (right); (**B**–**D**) interface of the modeled HK receiver domain surfaces colored by different attributes: (**B**) conservation; (**C**) hydrophobicity; (**D**) electrostatic potential.

**Figure 8 ijms-20-02096-f008:**
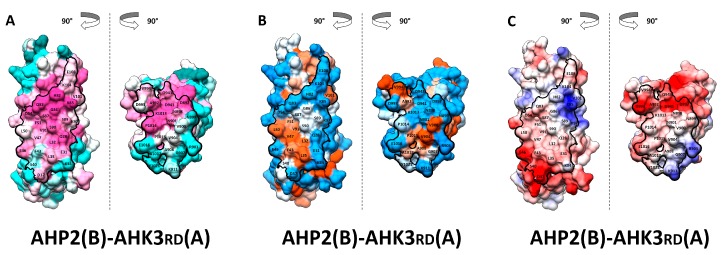
Properties of AHK3rd–AHP2 interface. (**A**) conservation; (**B**) hydrophobicity; (**C**) electrostatic potential. Color legends are the same as in Figure 2.

**Figure 9 ijms-20-02096-f009:**
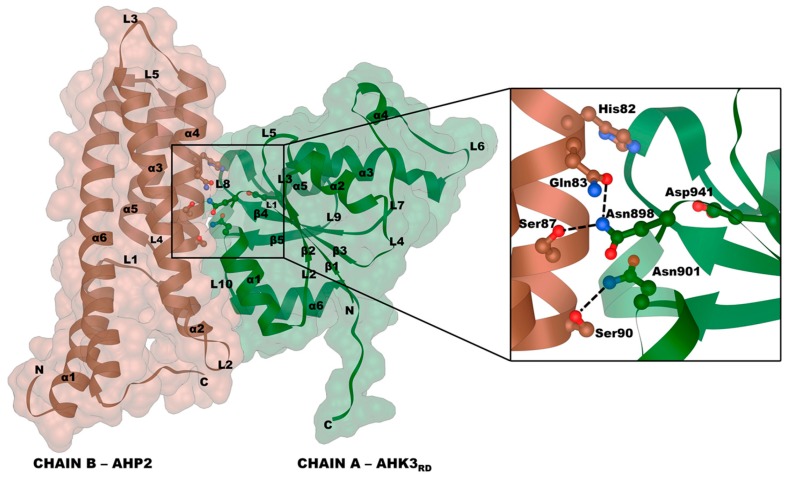
Interactions formed by hot spots in the interface of AHK3rd–AHP2 complex.

**Figure 10 ijms-20-02096-f010:**
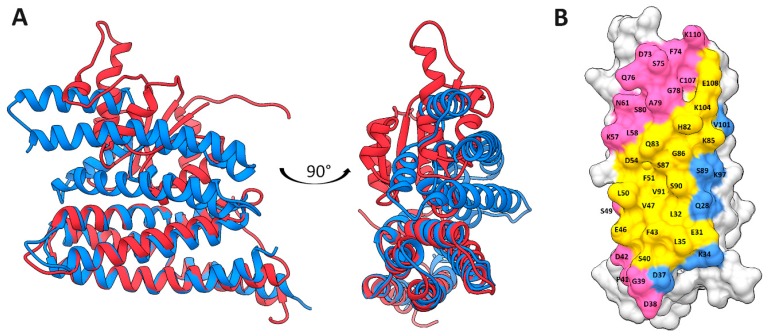
Comparison of AHK3rd–AHP2 complex and AHP2 homodimer structures. (**A**) blue, homology model of AHP2 dimer; red, homology model of AHK3–AHP2 complex; (**B**) crossing of the AHP2 interfaces: blue, residues involved only in AHP–AHKrd interaction; magenta, residues involved only in AHP–AHP interaction; yellow, common residues for both types of interaction.

**Figure 11 ijms-20-02096-f011:**
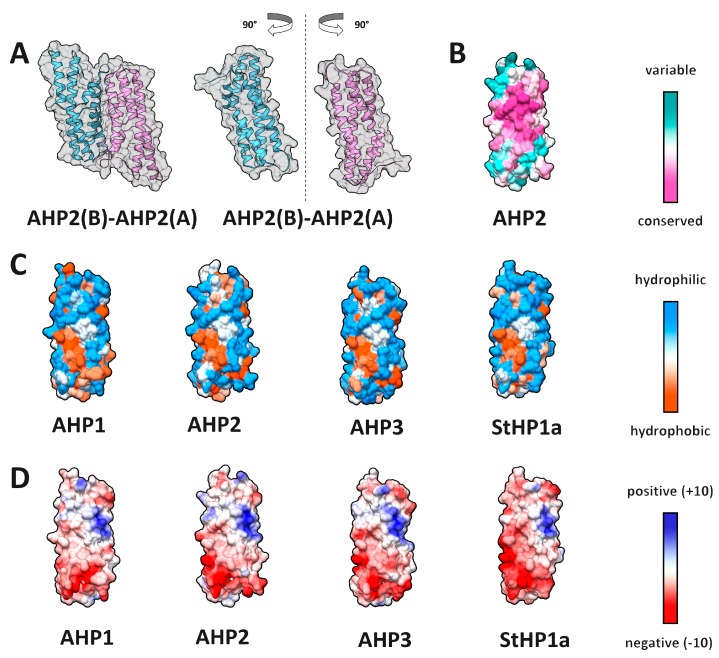
Properties of HPt interaction interfaces. (**A**) overall view of AHP2 homodimer (left) and its rotated subunits showing interface side (right); (**B**–**D**) interface side of the modeled HPt surfaces colored by different attributes; (**B**) conservation; (**C**) hydrophobicity; (**D**) electrostatic potential.

**Figure 12 ijms-20-02096-f012:**
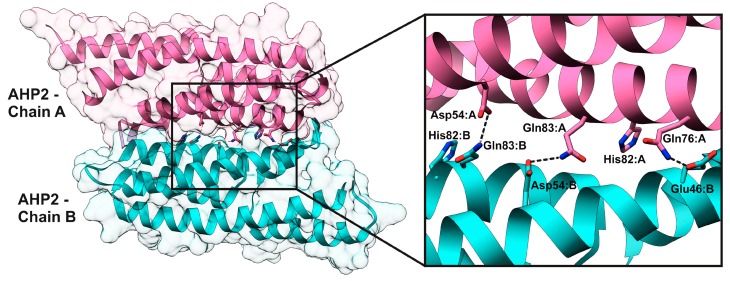
Hot spot interactions in the AHP2–AHP2 homodimer.

**Figure 13 ijms-20-02096-f013:**
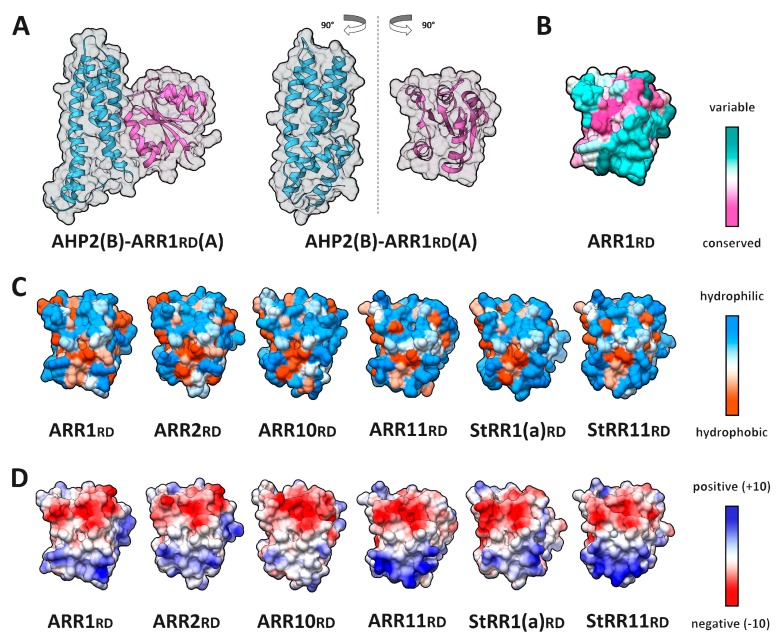
Properties of interaction interfaces of response regulators (RRs) receiver domains. (**A**) overall view of ARR1rd–AHP2 complex (left) and its rotated subunits showing interface side (right); (**B**–**D**) interface side of the modeled RR receiver domain surfaces colored by different attributes: (**B**) conservation; (**C**) hydrophobicity; (**D**) electrostatic potential.

**Table 1 ijms-20-02096-t001:** Phosphorylation and Mg^2+^ impacts on CK receptor(rd)–HPt interaction according to Prodigy server calculations.

Protein–Protein Complex	ΔG (kcal/mol)	ΔG (kJ/mol)	K_d_ (μM) at 25 °C
AHK3rd–AHP2_Apo	−8	−33.47	1.3
AHK3rd–AHP2_Mg^2+^	−8.5	−35.56	0.6
AHK3rd(**P**) –AHP2_Mg^2+^	−8.8	−36.82	0.35
AHK3rd–AHP2(**P**)_Mg^2+^	−7.9	−33.05	1.5
AHK3rd(**P**)–AHP2(**P**)_Mg^2+^	−8.1	−33.89	1.1
AHK3rd(**D941E**)–AHP2_Mg^2+^	−8.5	−35.56	0.54
AHK3rd–AHP2(**H82E**)_Mg^2+^	−8.4	−35.15	0.72

Bold indicates a phosphorylated (**P**) state of a protein or mimics (**D941E**, **H82E**) such a state.

**Table 2 ijms-20-02096-t002:** Templates used in homology modeling of protein complexes. More detailed data are in the Appendix A.

Modeled Structure		Template Structure		
Complex	Chain/Counterpart	PDB ID/chain ID	Protein Name	Reference
Sensory module dimers	chains AB	3T4L/AB	AHK4	[4]
HisKA domain dimers	chains AB	4MT8/AB	ERS1	[6]
HKrd–HPt complexes	HKrd (chainA)	3MMN/A	CKI1	[17]
		4EUK/A	CKI2(AHK5)	[19]
	HPt (chain B)	4EUK/B	AHP1	[19]
HPt–HPt dimers	chains AB	1YVI/AB	OsHP1	[23,24]
RRrd–HPt complexes	RRrd (chain A)	1CHN/A	CheY	[38]
		4EUK A	CKI2(AHK5)	[19]
	HPt (chain B)	4EUK B	AHP1	[19]

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
