# Peer review of "Modeling of Protein–Protein Interactions in Cytokinin Signal Transduction"

_ijms, 2019, doi:10.3390/ijms20092096_

Round 1
Reviewer 1 Report
This is an interesting piece of work that deals with the protein-protein interactions in cytokinin signal transduction using modeling. The authors used, for this purpose, two different plant systems, arabidopsis and potato. The paper is well-written, with a nice introduction that includes critical information to understand the main subject of the paper. The Results and Discussion section is nicely presented with the help of proper Figures, which include all the information needed to describe the main discoveries of the paper.
I have only a minor concern regarding with the conclusions and future lines of research:
Line 118-121. This paragraph could be part of the conclusion, as there are some results and recommendations that will fit into the general conclusion at the end of the paper.
It will also interesting if the authors could include within the Conclusions section future needs to follow up with the research and a wider description of the biological consequences of this research on the crop production area, more in particular, the case of potato.
Author Response
Reviewer 1. We are grateful to the reviewer for the benevolent evaluation of our work and valuable suggestions. Answers to specific comments follow.
1. C: Line 118-121. This paragraph could be part of the conclusion, as there are some results and recommendations that will fit into the general conclusion at the end of the paper.
A: This paragraph has been a bit modified and partly moved to the Conclusion part.
2. C: It will also interesting if the authors could include within the Conclusions section future needs to follow up with the research and a wider description of the biological consequences of this research on the crop production area, more in particular, the case of potato.
A: A fragment about the possible biological significance of the study and future prospects together with an additional reference has been added to the end of the Conclusion part.
Reviewer 2 Report
The authors provide a comprehensive study on molecular modeling of protein-protein interactions of the multistep phosphorelay system in cytokinin signaling. The CK-signaling proteins' structure from Arabidopsis and potato were modeled. Several "hot spots" amino acids were predicted to play important roles in dimerization, interaction, and phosphorylation across HK-HP-RR. The model results can be served as a potential regulatory mechanism of MSP.
Here are a few questions.
How conserved of those "hot spots" that identified in this study across the gene families in Arabidopsis, potato, and rice?
The affinity of AHK3-AHP2 interaction was increased in the presence of both Mg2+ and (P) on AHK3. Whether the interaction of AHP -AHP dimers or HP-RR can also be affected by the states of Mg2+ and (P)? Which has a lower affinity?
Line 504, 528, 531: The phosphorylation mimicking mutation in AHP2 is H82D or H82E (Table1)?
Line 487-488: Please consistent with the amino acid symbol with 1 or 3 letters.
Author Response
Reviewer 2. We thank the reviewer for well-minded consideration of our paper, a clear interest in our work and useful recommendations. Our answers (A) to reviewer's comments (C) are below.
1. C: How conserved of those "hot spots" that identified in this study across the gene families in Arabidopsis, potato, and rice?
A: Most of the determined hot spots are highly conserved. For example, N898, N901 and K1013 (Receptor’s receiver domain hot spots, according to AHK3 numeration) are conserved positions for all Arabidopsis, potato and rice CK receptors. N901 can be replaced by serine in CKI1, but this histidine kinase is not a CK receptor. In one of our previous articles we published a multiple alignment of all full sized CK receptors of Arabidopsis, potato and rice (Fig. 3 and Fig. S6, [8]). In the present paper we used ConSurf calculations to determine a conservation of all interfaced residues. ConSurf scores with reference to a hot spot status of interfaced amino acids are demonstrated in supplementary tables: Table S6 for sensor module dimer, Table S10 for HisKA domain dimer, Table S12 for AHK3rd–AHP2 complex and Table S15 for AHP2 homodimer.
2. C: The affinity of AHK3-AHP2 interaction was increased in the presence of both Mg2+ and (P) on AHK3. Whether the interaction of AHP -AHP dimers or HP-RR can also be affected by the states of Mg2+ and (P)? Which has a lower affinity?
A: We consider this modeling work as a some continuation of our precedent experimental study on CK receptors, phosphotransmitters and their interaction in MSP signal transduction [30]. Therefore when studying the effect of phosphorylation, we have focused primarily on the receptor-phosphotransmitter interaction which is a critical stage in CK signaling. Concerning other interactions, our preliminary data showed that the phosphorylation reduces the stability of HPt dimers, but these data need additional confirmation. Overall this kind of research is a separate widescale study which is a part of our plans for the future.
1. C: Line 504, 528, 531: The phosphorylation mimicking mutation in AHP2 is H82D or H82E (Table1)?
A: We thank the Reviewer for noticing this inaccuracy. H82E is correct. Typos have been fixed.
2. C: Line 487-488: Please consistent with the amino acid symbol with 1 or 3 letters.
A: Symbols were changed to the single letter.